# Health- or Environment-Focused Text Messages as a Potential Strategy to Increase Plant-Based Eating among Young Adults: An Exploratory Study

**DOI:** 10.3390/foods10123147

**Published:** 2021-12-19

**Authors:** Tze Joo Lim, Richard Nii Okine, Jonathan C. Kershaw

**Affiliations:** 1Department of Public and Allied Health, Bowling Green State University, Bowling Green, OH 43403, USA; tlim@bgsu.edu; 2Department of Mathematics and Statistics, Bowling Green State University, Bowling Green, OH 43403, USA; rnokine@bgsu.edu

**Keywords:** text message, plant-based diet, meat consumption, sustainability, moral satisfaction, self-efficacy, subjective norm, protein intake

## Abstract

Previous plant-based diet (PBD) adoption strategies have primarily focused on health rather than environmental rationale and meat reduction rather than plant-based protein promotion. In this study, we explored the effect of a theory-informed text-message intervention on dietary intentions and behaviors in young adult omnivores and the potential explanatory role of PBD beliefs, subjective norm, self-efficacy, moral norm, and health and environmental values. Participants completed baseline questionnaires and reported dietary intake before being randomly assigned to receive 2–3 health- or environment-focused text messages per week for eight weeks and then repeated baseline assessments. Although we did not see significant changes in meat or plant protein intake, we did observe a marked decrease in intentions to consume animal protein and a marginal increase in fruit and vegetable consumption intention. We identified subjective norms, self-efficacy, and moral satisfaction as the strongest predictors of changes in intention to consume animal or plant protein. Although few group differences were observed, those receiving environment-focused text messages experienced a greater change in values and were more likely to increase vegetable intake. Messages that improve sustainability awareness and provide practical adoption strategies may be part of an effective strategy to influence PBD intake among young adults.

## 1. Introduction

Adoption of plant-based diets (PBD) improves both health and environmental outcomes and aligns with recommendations stated in the Dietary Guidelines for Americans (DGA) [1,2]. PBD emphasize vegetables, fruits, whole grains, nuts, seeds, and their manufactured products and generally focus on reduction rather than elimination of food items from animal sources. However, for several decades, approximately two-thirds of American protein intake has come from animal foods [3,4,5,6]. At the same time, many populations continue to fall short of dietary targets despite introduction of the DGA in 1980 [1,2,7,8]. This relatively static state of diet quality suggests that health-focused strategies alone may be insufficient to stimulate meaningful changes. Considering the rising interest in environmental sustainability [9,10], environment-focused messaging may represent a novel strategy to complement existing efforts to improve diet quality.

### 1.1. Health and Environmental Effects of Plant-Based Diets

Plant foods are health-promoting and protective against some diseases [11,12]. Previous studies have consistently shown that a plant-based diet reduces the risk of cardiovascular disease-related mortality, most chronic diseases, type 2 diabetes, hypertension, some cancers, and obesity [11,12]. Plant foods also provide the only source of phytochemicals and fibers, which play a role in health promotion and disease prevention [12]. While animal foods are rich in essential nutrients, such as iron, zinc, vitamin D, calcium, and essential fatty acids [12], certain animal products may increase disease risk if consumed in excess. For example, some animal products contain high amounts of saturated fats and other compounds that have putative roles in progression of heart disease, cancer, and other diseases [12,13].

In addition to health benefits, PBD contribute to a number of environmental benefits. Non-vegetarian diets require 2.5 times more energy, 2.9 times more water, 13 times more fertilizer, and 1.4 times more pesticides than vegetarian diets [14]. The 2015 U.S. Dietary Guidelines Advisory Committee concluded that a diet higher in plant-based foods (whole grains, legumes, nuts, seeds, fruits, and vegetables) and lower in animal-based foods is better for health and the environment than the typical U.S. diet [15,16]. 

### 1.2. Facilitators and Barriers to Adoption of Plant-Based Diets

Although plant-based diets bring about many health and sustainability benefits, shifting dietary patterns is complex. Consumers’ attitudes toward and intake of PBD vary widely and are associated with differences in gender, income, age, race/ethnicity, and political leaning [17,18]. Consumers who have a positive attitude toward prosumerism, ethics, health, and naturalness are supportive of a transition to a plant-based diet, while social image and pleasure are barriers to a transition to a plant-based diet [19]. Consumers whose established diets consist of beans and soy products have a high preference for plant-based food due to health, weight maintenance, and natural concerns compared with those with high intakes of animal food [20]. Interestingly, those who are undergoing a dietary transition toward the consumption of plant proteins endorsed higher regard for health, natural concerns, price, sociability, and social image [20].

While interest in sustainable eating is growing, general confusion regarding its meaning and a lack of awareness of the need to consume more plant-based foods limit its more widespread adoption. For example, while most Americans have heard of PBD, many people associate sustainability with only tangentially related factors, such as organic or non-genetically engineered foods [10,21]. Furthermore, several knowledge-related (e.g., unaware of the need to consume PBD) and ability/access-related barriers to PBD consumption (e.g., cost, knowledge regarding what to buy or how to cook, and perceived effort to prepare) limit its adoption [22]. Thus, addressing the gap in awareness and abilities may encourage a shift to PBD. 

### 1.3. Text-Message Intervention as a Tool for Nutrition/Health Behavior Change

Due to the widespread ownership of mobile phones across gender, age, race, education level, income level, and geographical location [23], short-messaging service (SMS) represents a possible intervention to educate concerning PBD benefits and influence dietary behavior change. Indeed, text-messaging interventions have shown promising results in influencing health behavioral change, including increased knowledge and eating behavior among college students, higher adherence to dietary guideline recommendations in patients with coronary heart disease, increased healthy eating behaviors among university students, and decreased red meat consumption among young adults [24,25,26,27]. 

Despite the need to increase PBD and the potential of text messages to influence dietary knowledge and behaviors, few studies have investigated text messages as a strategy to encourage plant-protein consumption. Previous studies have focused on either increasing consumption of fruits and vegetables or decreasing consumption of red meat rather than PBD collectively, and few have explored the effect of environmental sustainability messaging [28]. As environmental impacts of diet are generally less well-known, sustainability messaging may be an effective strategy to influence diet [21,22]. Interestingly, courses focusing on the environmental impact of food have influenced food intake in young adults even more than health-focused courses [29,30]. Furthermore, sustainability messages have shown promise in influencing diet quality of young adults [31,32]. Considering the growing interest in environmental sustainability—especially among young adults [9,10]—this approach merits further investigation.

## 2. Theoretical Framework

Both the Health Belief Model (HBM) and the Theory of Planned Behavior (TPB) have been effective in promoting health behavior change [33,34,35]. The HBM is based on the desire to avoid sickness or get well if already ill and the belief that a specific health action will prevent or cure the sickness [36]. Thus, an individual’s course of action depends on how they perceive the benefits of adopting the behavior and the consequences of failure to change. It follows that messages targeting these beliefs may influence the desired behavior. The HBM identifies perceived benefits, perceived susceptibility, perceived severity, cues to action (factors that remind individuals to adopt healthy behavior), and self-efficacy as predictors of health behaviors [36]. Constructs from HBM have informed effective change across different behaviors and media, including behaviors associated with diet-related stomach cancer via a text-messaging intervention and osteoporosis prevention via an educational program [37,38]. The Theory of Planned Behavior (TPB), previously known as the Theory of Reasoned Action, predicts an individual’s intention to engage in a behavior [39]. It states that behavioral achievement is the result of attitudes (which are formed, at least in part, from underlying beliefs about the behavior), perceived behavioral control (similar to self-efficacy in that both measure belief in one’s control over the behavior, but operationalized slightly differently [40]), and subjective norms (belief whether the behavior is approved or disapproved by the people of importance to the person) [39]. In addition to the traditional TPB constructs, others have proposed additional factors that uniquely increase the prediction of behavior. For example, several authors have proposed that moral norm or moral satisfaction adds unique predictive power beyond TPB constructs in explaining food intake intentions [41,42]. Of note, text message interventions are generally more influential when multiple behavior-change techniques are incorporated [33].

### The Current Study

Targeting theoretical constructs from HBM and TPB, we designed and compared the effect of two parallel text-message interventions (health-focused and environment-focused) on influencing dietary intention and behavior, with a focus on protein selection. Specifically, the intervention contained messages describing health or environmental benefits of PBD, potential health or environmental consequences of animal-based diets, practical strategies to increase plant-based eating (including links to supportive websites in each message), and information about the increasing popularity of PBD among Americans. Our first objective was to identify the effect of the 8-week text-message intervention (health-focused or environmental focused) on potential dietary predictors (i.e., values, PBD beliefs, subjective norm, self-efficacy, and moral satisfaction), intentions, and actual intake.

In addition to exploring the effectiveness of the intervention itself, we also examined relationships among dietary predictors, intentions, and actual intake. Because environmental, ethical, and health values may influence intention and moderate the effect of other predictors [43,44,45], we explored potential direct and indirect roles of values on intentions and behaviors. Furthermore, due to the documented role of perceived benefits, perceived susceptibility, perceived severity, subjective norms, and moral satisfaction in explaining dietary and health-related behavior changes [34,35,37,45,46,47,48], we explored the extent by which these predictors explained changes in intentions and intake. 

## 3. Methodology

This study consisted of four phases: recruiting and screening, baseline data collection (one survey and two dietary assessments), an eight-week text-message intervention, and post-intervention data collection (one survey and two dietary assessments). 

### 3.1. Participants

Young adults between 18 to 26 years old within the United States were recruited via online platforms, including Facebook, Amazon mechanical Turk, and FindParticipants.com. To qualify, participants indicated that they could understand and read English without assistance, owned a mobile phone, were an active user of SMS, were responsible for at least half of their meal choices (purchases, preparation etc.), and consumed both plant-based and animal-based foods. Participants were excluded if they indicated they followed a vegan or vegetarian diet (e.g., lacto-vegetarian, ovo-vegetarian, lacto-ovo-vegetarian, or pescatarian), were planning to change their diet in the next 3 months (e.g., trying to lose weight), or had a history of eating disorder(s). Using standard error estimates for protein intake among males and females ages 20–29 [49], we calculated a sample size of at least 68 participants per group would be required to detect a difference of one serving of protein (approximately 25 g) with 80% power at an alpha of 0.05. 

A sample of 505 participants met the initial screening criteria. All of the interested participants provided informed consent before participating in the study. Of the initial participants, 201 successfully completed the baseline survey and diet assessments. Following completion of baseline surveys, participants were randomly assigned to one of two text message interventions (described below). Of these participants, 159 completed the study (see Figure 1). Participants were awarded an e-gift card both after completing the initial surveys and after completing the final surveys. 

### 3.2. Survey Instruments

#### 3.2.1. Dietary Assessment

Following study enrollment, participants were emailed an invitation to complete two dietary assessment surveys on two unannounced days (one weekday, one weekend day). Dietary intake data were collected and analyzed using the Automated Self-Administered 24-h (ASA24) Dietary Assessment Tool, version 2020, developed by the National Cancer Institute (Bethesda, MD, USA), which has been validated for use in large-scale nutrition research [50,51]. The ASA24 guides participants through a series of questions to assess the foods and beverages that were consumed the previous day. Participants who failed to complete the assessment the day it was received were emailed an unannounced invitation on a later day. Participants also completed two additional ASA24 dietary assessments immediately following the text-message intervention. For the purpose of this study, animal-based foods refer to meat (beef, pork, goat, lamb, and venison), poultry (chicken, turkey, and bird), seafood (fish and shellfish), eggs, dairy, and their manufactured products. Plant-based foods refer to fruits, vegetables (dark-green vegetables, red and orange vegetables, legumes, starchy vegetables, and other vegetables), whole grains, nuts, seeds, and their manufactured products. 

#### 3.2.2. Study Questionnaire

The survey questionnaire was administered online via survey software. Participants answered questions regarding their personal values (health and green consumer values), health- and environment-focused questions concerning PBD beliefs (perceived benefits of consumption; perceived severity of and susceptibility to consequences of not consuming), self-efficacy, subjective norms, and moral satisfaction using 7-point Likert scales anchored by strongly disagree and strongly agree. Participants also stated their intention to consume plant-based proteins, fruits and vegetables, and animal-based protein sources using a 100-point sliding scale. Survey items were adapted from previously established valid and reliable instruments [34,35,52,53] (Appendix A, Table A1). To assess content validity, we ensured that the various items covered the construct of interest. Internal consistency of survey items was assessed using Cronbach’s alpha as a measure of reliability [54]. Cronbach’s alpha coefficient can range from 0.0 to 1.0. A Cronbach’s alpha close to 1.0 indicates that the item is considered to have a high internal consistency reliability, above 0.8 is considered good, and 0.7 is considered acceptable [55]. Cronbach’s alpha for all constructs was above 0.7, with the vast majority exceeding 0.890 (Appendix A, Table A1).

One attention check question was also included; data from participants who incorrectly answered the attention check question were removed. General participant characteristics, including age, height, weight, gender, ethnicity, educational level, annual household income, biggest meal of the day, and time zone of place of residence, was also collected. Upon completion of the final dietary assessments, participants were emailed the same survey previously described to collect post-intervention data, with the exclusion of demographic questions.

### 3.3. Text Message Intervention

After participants completed baseline data collection, they were randomly divided into two treatment groups: Health Message (HM) and Environment Message (EM). Minor modifications were made so that the groups were comparable in terms of group size, education, income, age, and gender at baseline. Participants then received an initial text message to confirm their participation in the study. Once participants confirmed receipt of the introductory text message, they began to receive the study text messages. Thirty-two messages were developed: 16 health-focused and 16 environment-focused. The behavioral theory constructs served as guide in developing the SMS. The text messages for each group were comparable in content and structure, with only slight variation in the words related to the corresponding intervention. Text messages were screened for understandability and effectiveness and modified accordingly. Due to the small but significant effect of supplementary materials on text-message effectiveness [33], we also included links to websites to aid participants in PBD implementation. The complete list of SMS can be found in Appendix A, Table A2.

Participants received 2–3 SMS per week for eight consecutive weeks on Tuesdays (a weekday) and Fridays (preceding the weekend) via Textedly (SMS marketing software; textedly.com). The participants received the SMS at either 11:30 am or 4:30 pm depending on their largest meal as indicated in the pre-survey, as previous studies indicated that SMS are most useful for behavior change when received at high-risk situations [24,56]. The majority of the SMS delivery was unidirectional except for a biweekly SMS asking participants to text back 1 (not at all), 2 (maybe), or 3 (likely) to “How likely are you to focus on eating plant-based protein next week?” and “How likely are you to focus on eating more fruit and vegetables next week?”

### 3.4. Statistical Analysis

Comparability of participant characteristics between groups was assessed using the chi-square statistic. To explore the effects of the text-message interventions on dietary predictors, intentions, and behaviors, we conducted paired-samples *t*-tests using baseline and post-intervention responses. To explore potential differences between those assigned to health-focused vs. environment-focused text messages, an independent samples *t*-test of changes in outcomes (post data–baseline data) between the two groups was conducted. Correlations between changes in outcomes and education and income were calculated using the Pearson’s correlation. Gender differences were compared using an independent samples *t*-test of changes in outcomes. Due to a small sample size, the “other” gender category was not included in the analysis. 

Considering the exploratory nature of this study, we investigated the effect of the collective text-message intervention on relationships among predictor variables and outcomes using two analyses. First, we examined whether baseline data predicted changes in outcome variables by regressing the post-intervention outcome variable on the baseline predictor variable while controlling for the baseline value of the outcome variable, as suggested by Cole and Maxwell for mediation analysis of half-longitudinal data [57]. Group (health vs. environment) was also included in the model to identify possible differences between HM and EM. As a second analysis, we used the Pearson’s correlation coefficient to explore relationships among changes in predictors, intentions, and intake.

In all analyses, no adjustments were made for multiple comparisons. For analyses including predictor variables, only data from the survey instrument that corresponded with the intervention were included (e.g., health values were used as “values” for the HM, and green consumer values were used as “values” for the EM). All analyses were conducted using SPSS version 26.

## 4. Results

The groups were comparable in terms of group size, gender, education, and income, as shown in Table 1.

Due to the exploratory nature of this study, a number of comparisons and analyses were conducted, and thus, results should be interpreted within context of the study’s objectives. In the subsequent results and discussion, we use “marginally significant” when *p* < 0.05 and “significant” to describe results where *p* < 0.01. Greater emphasis is given to significant results and/or consistent trends that were marginally significant across multiple variables.

### 4.1. Effect of SMS on Predictor Variables and Dietary Outcomes

Our first objective was to explore and compare the collective and individual effect of the health-focused and environmental-focused eight-week text-message interventions on PBD beliefs, subjective norms, self-efficacy, moral satisfaction, dietary intentions, and actual intake of protein foods and fruits and vegetables. Because no group differences (*p* < 0.05) were detected when directly comparing changes in variables using an independent samples *t*-test (data not shown), the data were combined for a single analysis, and results for each group are displayed for individual comparison (Table 2). Following the text message intervention, we observed a significant increase in moral satisfaction and perceived benefits of PBD and a marginally significant increase in perceived susceptibility and self-efficacy. Although we initially assumed values to be a stable state [58], we observed a marginally significant increase in values, which appears to be driven largely by a significant increase in green consumer values only within the group that received the environment-focused text messages; no differences were observed in health values in either group. Upon further comparison of differences in each group, only perceived susceptibility was marginally higher in the group receiving the health messages. 

Consistent with our hypotheses, intentions to consume animal-based protein sources significantly decreased while intentions to consume fruits and vegetables marginally increased following the intervention. Although the study was originally designed with the goal to increase plant-based protein intake, the increase in intentions to consume plant protein foods did not reach statistical significance. Despite intentions to increase fruit and vegetable intake and decrease animal-based protein intake, significant changes in actual food intake servings were only observed for eggs. Interestingly, a marginal increase in vegetable intake was observed only in the group that received the environment-focused text messages. We also assessed whether changes differed by gender, education, and income. Higher education marginally correlated with increased meat, poultry, and seafood intake. Income was positively correlated with changes in legume intake and marginally negatively correlated with changes in intention to consume animal foods. Females marginally consumed more seafood, while males marginally consumed more eggs (Table 3).

### 4.2. Relationships among Predictor and Outcome Variables

#### 4.2.1. Relationships among Baseline Values and Other Predictor Variables

Due to the direct effect of personal values on food behaviors and mediators of dietary intention [42,45], we next explored the extent by which values influenced dietary predictors, intentions, and behaviors (Table 4). Higher baseline values consistently predicted significantly larger SMS-induced increases in perceived susceptibility, perceived severity, and moral satisfaction and marginal increases in subjective norms and self-efficacy. Interestingly, the largest effect size was observed for perceived severity of failure to consume PBD, which was more pronounced in the group receiving health-focused messages. While baseline values did not predict a change in perceived benefits, changes in values were positively correlated with changes in perceived benefits. Changes in subjective norms also positively marginally correlated with changes in values.

#### 4.2.2. Relationships among Dietary Intentions and Predictor Variables

We consistently observed the anticipated relationship between predictor variables and intentions to consume animal (consistently negative) and plant protein (consistently positive) sources in nearly all measured variables (Table 4). Increases in intentions to consume plant-based protein were significantly predicted by subjective norm and self-efficacy; marginally predicted by baseline susceptibility, severity, and moral satisfaction; and marginally correlated with increases in perceived benefits. Decreases in intention to consume animal protein sources were predicted by higher baseline subjective norm, moral satisfaction, and values and correlated with increases in perceived severity and self-efficacy. Of note, no variables predicted changes in intention to consume fruits and vegetables. 

#### 4.2.3. Relationships among Actual Intake and Predictor Variables

Higher baseline perceived benefits of PBD and self-efficacy marginally predicted an increased effect of the SMS intervention on plant protein intake (Table 4). Contrary to our hypothesis, higher baseline perceived susceptibility, subjective norm, and values predicted an increase in the combined category of meat, poultry, and seafood intake. Contrastingly, when meat was analyzed separately, opposite trends were observed: higher moral satisfaction marginally predicted lower meat intake, and meat intake marginally decreased as self-efficacy increased. Although we did not observe strong predictive effects of explanatory variables on fruit or vegetable intake, we did observe a small group effect for values, suggesting that environmental-focused text messages have a greater influence on vegetable intake for those with higher environmental values than do health-focused messages on those with higher health values, consistent with our observation of a greater overall vegetable intake in the environment group (Table 2).

## 5. Discussion

In this study, we used a TPB- and HBM-informed approach to explore the effect of a health-focused and environment-focused text-message intervention on dietary intake and intention, with a focus on protein source. We also explored the possible predictive role of values, PBD beliefs (i.e., perceived PBD benefits, perceived severity of and susceptibility to consequences of not consuming PBD), subjective norms, self-efficacy, and moral satisfaction. 

### 5.1. Implications for Plant Protein

Although intentions and actual intake of plant-protein sources did not change as a result of the intervention, all measured predictors significantly or marginally explained increases in plant protein intention and/or intake. Our observation of relatively stronger effect sizes for subjective norms and self-efficacy extends previous findings: while others identified a lack of knowledge and awareness of others’ expectations as potential PBD barriers [22], we identified self-efficacy and subjective norms as a potential target to improve intentions. When combined with other strategies, messaging that increases awareness of and practical strategies to incorporate plant-based protein sources may influence PBD adoption.

As acceptance of alternative proteins is affected by food neophobia, the fear of trying new or unfamiliar foods, lack of familiarity with plant-based protein sources may be one explanation for our failure to detect an increase in plant-protein intake [59,60]. Indeed, food neophobia is negatively associated with liking beans and legumes [60]. The unfamiliarity of PBD and plant proteins may have induced reluctance to incorporate plant protein foods and adoption of PBD. Perceived taste is also a barrier to PBD adoption [22], and thus, future studies should explore the impact of messaging on taste perception. Because taste and familiarity are indispensable determinants of acceptance for healthy and sustainable foods [61], interventions that target these attributes may be necessary to increase intake for some populations. 

### 5.2. Implications for Animal Protein

The text-message intervention resulted in a meaningful and significant decrease in intentions to consume animal foods. This decrease was explained in part by baseline moral satisfaction, subjective norms, and values and correlated with increases in self-efficacy. Consistent with our findings in plant-based protein intention, our observation that actual meat intake decreased as self-efficacy increased suggests that messages targeting abilities to prepare and consume PBD may be an effective strategy to influence protein selection. The effectiveness of messages promoting the normalcy of reducing meat consumption is supported by our data and others’ findings that normative meat-eating statements lowered self-reported interest in meat-eating and actual orders of a meatless lunch [46]. Furthermore, our data suggest potential strategies for targeting meat intake among individuals with certain values or beliefs: PBD messages with an ethical focus may resonate with the increasing number of people that consider animal welfare a moral issue [62], while health messages may be more effective for those that perceive the health severity of animal-based diets, consistent with others’ observations of greater meat discouragement among those that believed meat was bad for health but not among those that believed it was bad for the environment [17].

Our observation that meat intake but not poultry and seafood intake changed in the expected direction may have several possible explanations. Firstly, these observations support the efficacy of the text-message intervention, as meat was defined for participants as distinct from poultry or seafood, and several text messages mentioned meat specifically, but none mentioned poultry or seafood. Others’ finding that over one-third of meat-reducers report increasing poultry consumption may also explain our observation [18]. We hypothesize that our observed increase in egg consumption may have a similar explanation: participants may have increased non-meat protein sources in response to the text messages. 

### 5.3. Health vs. Environmental Messaging

Although we observed few differences between the EM and HM groups, several observations suggest a small but unique role of environment-focused messages. Firstly, we observed that green consumer values significantly increased only in the EM group and predicted increased vegetable intake to a greater extent than health values in the HM group. Importantly, higher vegetable intake was observed only in the EM group. Furthermore, baseline values consistently explained changes in meaningful predictors of dietary intention and behavior. Low knowledge and awareness of PBD [21], low involvement in sustainable eating compared with healthy eating [63], and lack of evidence regarding poor nutrition knowledge as a predictive factor in healthy eating [64,65] together suggest that environmental (but not health) education may present an opportunity: increasing knowledge about environmental effects of a PBD may alter green values, which in turn may influence dietary choices.

However, we note that no direct group difference was observed for any of the measured outcomes, perhaps due to relatively lower statistical power. Thus, the content of the text message (i.e., health or environment) may be less influential than other factors, such as targeting certain beliefs, attitudes, or self-efficacy. While environment-focused messaging alone would likely be insufficient to influence PBD adoption, we suggest that sustainability messaging may be part an overall strategy to influence plant-based eating among young adults. Furthermore, considering that those involved in sustainable eating tend to also be highly involved in healthy eating (but not necessarily the opposite) and the considerable overlap between perception of healthy, sustainable, and plant-based diets, sustainability messaging would likely be compatible with existing guidelines for healthy eating [63]. More research is needed to understand the unique effect of health vs. environmental messages.

### 5.4. Limitations, Strengths and Future Directions

Despite the convenience, non-disruptive nature, and low cost of text-message interventions, further research is needed to understand how factors, such as frequency, length of intervention, timing of delivery, tone, and duration of the effect, can be optimized [66,67]. Furthermore, knowing whether participants actually read and understand the messages is a challenge, similar to other education interventions, such as emails or classes [24]. Evidence from systematic reviews suggest that although text messages can be effective for promoting health behaviors, further research is needed to optimize implementation [66]. Potential strategies to optimize text-message interventions include greater interactivity and engagement strategies, such as accessibility to an advisor and text message tone [33,68]. 

Interpretation of our results should be considered in context of several limitations. Although efforts were taken to balance gender representation in the study sample, greater attrition among males resulted in a disproportionate number of female representation in our sample. We also recognize that most of the effect sizes we observed were relatively small and that long-term persistence of our observations require further study. Although we incorporated a number of relevant behavioral constructs to explain dietary intentions and behavior, the presence of other possible explanatory factors should also be considered, such as self-regulation [28] and self-identity [42]. Importantly, the exploratory nature of the data analysis must also be considered when interpreting our findings. Additional a priori investigations and full mediation analysis are necessary to confirm the results of the current study. 

Despite its limitations, this study also provides several novel contributions. While previous studies have focused primarily on decreasing animal-protein intake and increasing fruit and vegetable intake [28], our focus on plant protein adds needed insights, especially considering the increasing popularity of plant-based protein [69]. Of note, we collected dietary intake using a comprehensive and validated tool, thus strengthening our conclusions about the impact of the intervention on actual intake. Importantly, vegans and vegetarians were excluded from the study, and thus, our findings are better applied to populations currently consuming animal-based protein sources. 

## 6. Conclusions

In this study, we reveal a possible role of text-messages in decreasing intention to consume animal protein sources and increasing intention to consume fruits and vegetables. However, using a validated dietary assessment tool, we detected significant dietary changes only for egg intake. We also identified several conditions when messaging interventions may be more effective in influencing protein-source intake: individuals with relatively higher self-efficacy regarding their abilities to consume PBD, a feeling of the moral rightness of PBD, and a sense of the subjective normalcy of PBD consumption may be more influenced by messaging campaigns to encourage PBD adoption. While we did not observe a consistent advantage of an environment vs. health-focused messaging campaign, our findings of altered green consumer values and vegetable intake in the EM group suggest that environment-focused messages may be compatible with existing strategies to encourage healthier and more environmentally friendly dietary patterns among young adults.

## Figures and Tables

**Figure 1 foods-10-03147-f001:**
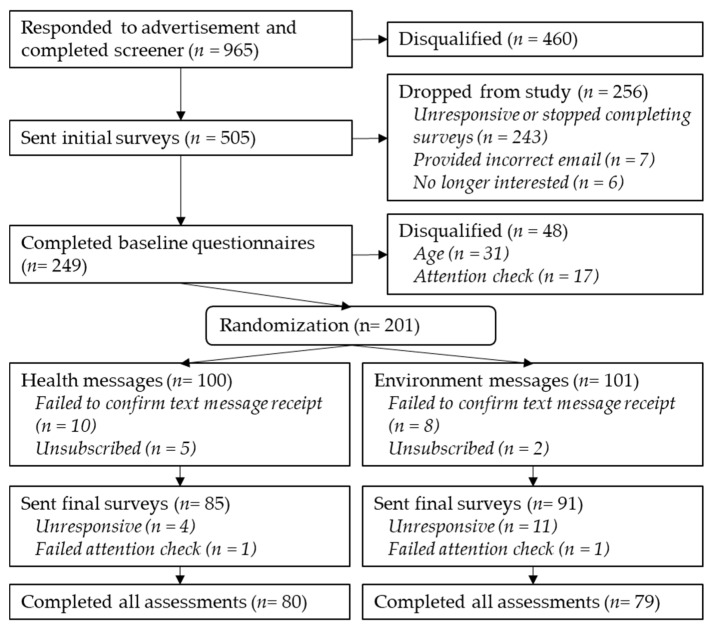
Flowchart of study participants.

**Table 1 foods-10-03147-t001:** Participant characteristics.

Demographic	Total	HM	EM	Pearson χ^2^ Sig.
**Gender**				
Female	107	55	52	0.743
Male	49	23	26	
Other	3	2	1	
**Education Levels**				
Less than high school degree	0	0	0	0.244
High school graduate (high school diploma or equivalent including GED)	9	4	5	
Some college but no degree	39	14	25	
Associate degree in college (2-year)	9	6	3	
Bachelor’s degree in college (4-year)	50	28	22	
Some graduate (Master’s, doctorate) or professional (MD, JD, etc.)	30	14	16	
Graduate or professional degree	22	14	8	
**Annual Income**				
$0–$24,999	33	11	22	0.296
$25,000–$49,999	42	22	20	
$50,000–74,999	25	13	12	
$75,000–$99,999	19	12	7	
$100,000–$149,000	24	12	12	
$150,000 or more	16	10	6	
**Total**	**159**	**80**	**79**	

**Table 2 foods-10-03147-t002:** Mean values and changes in dietary predictors, intentions, intakes, and intentions following the intervention. Bold values indicate *p* < 0.05. Dashed lines indicate *p* < 0.001. Food intake is reported as number of servings.

	Total				Health Messages (HM)				Environment Messages (EM)			
	T1 (SD)	T2 (SD)	Δ	*p*	T1 (SD)	T2 (SD)	Δ	*p*	T1 (SD)	T2 (SD)	Δ	*p*
**Values**	5.32 (1.03)	5.44 (0.93)	**0.11**	**0.048**	5.72 (0.82)	5.70 (0.87)	−0.02	0.743	4.92 (1.08)	5.17 (0.92)	**0.25**	**0.007**
**Health value**	5.69 (0.84)	5.69 (0.85)	0.00	0.924	5.72 (1.05)	5.70 (0.87)	−0.02	0.743	5.66 (0.86)	5.69 (0.83)	0.03	0.654
**Green consumer value**	5.02 (1.07)	5.18 (1.04)	**0.16**	**0.007**	5.12 (0.82)	5.19 (1.16)	0.07	0.362	4.92 (1.08)	5.17 (0.92)	**0.25**	**0.007**
**Moral** **satisfaction**	4.41 (1.52)	4.81 (1.40)	**0.39**	**---**	4.35 (1.56)	4.74 (1.47)	**0.39**	**0.009**	4.47 (1.47)	4.87 (1.34)	**0.40**	**0.002**
**Perceived** **benefits**	5.24 (1.28)	5.54 (1.26)	**0.30**	**---**	5.20 (1.34)	5.53 (1.36)	**0.33**	**0.010**	5.29 (1.23)	5.56 (1.15)	**0.27**	**0.011**
**Perceived** **susceptibility**	4.13 (1.47)	4.38 (1.52)	**0.26**	**0.017**	4.22 (1.51)	4.55 (1.51)	**0.33**	**0.028**	4.03 (1.43)	4.21 (1.52)	0.18	0.241
**Perceived** **severity**	5.36 (1.18)	5.49 (1.20)	0.12	0.111	6.00 (0.86)	6.08 (0.97)	0.08	0.387	4.73 (1.12)	4.89 (1.11)	0.16	0.180
**Subjective norms**	4.34 (1.54)	4.51 (1.50)	0.17	0.089	4.43 (1.57)	4.67 (1.46)	0.25	0.103	4.25 (1.51)	4.34 (1.52)	0.09	0.490
**Self-efficacy**	5.40 (1.15)	5.56 (1.11)	**0.16**	**0.031**	5.53 (1.15)	5.68 (1.10)	0.15	0.185	5.26 (1.15)	5.43 (1.12)	0.17	0.080
**Total protein foods**	5.44 (3.40)	5.74 (3.98)	0.30	0.341	5.03 (3.09)	5.65 (3.77)	0.62	0.182	5.85 (3.66)	5.84 (4.21)	−0.02	0.969
**Meat, poultry, seafood**	4.06 (3.07)	4.29 (3.54)	0.23	0.431	3.81 (2.67)	4.20 (3.40)	0.39	0.328	4.31 (3.42)	4.39 (3.71)	0.07	0.868
**Meat**	1.14 (1.85)	1.13 (1.78)	−0.02	0.936	1.13 (1.72)	1.03 (1.49)	−0.10	0.698	1.16 (2.00)	1.22 (2.04)	0.07	0.801
**Poultry**	1.64 (1.98)	1.74 (2.26)	0.10	0.644	1.54 (1.81)	1.82 (2.40)	0.28	0.292	1.75 (2.15)	1.66 (2.12)	−0.09	0.794
**Seafood**	0.98 (1.22)	0.72 (1.87)	−0.25	0.146	0.89 (1.22)	0.58 (1.29)	−0.30	0.118	1.07 (1.22)	0.87 (2.32)	−0.20	0.486
**Eggs**	0.26 (0.64)	0.66 (0.95)	**0.40**	**---**	0.21 (0.57)	0.62 (0.79)	**0.41**	**---**	0.31 (0.71)	0.70 (1.10)	**0.39**	**0.012**
**Nuts and seeds**	0.54 (0.94)	0.50 (0.90)	−0.04	0.622	0.48 (0.70)	0.53 (0.96)	0.05	0.640	0.60 (1.13)	0.47 (0.85)	−0.12	0.296
**Legumes**	0.45 (0.89)	0.50 (0.97)	0.05	0.593	0.46 (0.88)	0.49 (0.93)	0.03	0.839	0.43 (0.90)	0.51 (1.02)	0.08	0.594
**Soy**	0.18 (0.63)	0.29 (0.77)	0.12	0.145	0.18 (0.69)	0.30 (0.77)	0.12	0.283	0.17 (0.56)	0.28 (0.78)	0.11	0.328
**Total dairy**	1.46 (1.41)	1.31 (1.24)	−0.15	0.234	1.55 (1.69)	1.45 (1.42)	−0.10	0.592	1.36 (1.06)	1.17 (1.01)	−0.19	0.233
**Total fruits**	0.80 (0.93)	0.82 (0.96)	0.02	0.801	0.79 (0.97)	0.82 (1.05)	0.03	0.793	0.80 (0.89)	0.81 (0.88)	0.01	0.934
**Total vegetables**	1.43 (0.98)	1.61 (1.05)	0.17	0.068	1.50 (1.06)	1.50 (0.93)	0.00	0.999	1.37 (0.90)	1.72 (1.16)	**0.35**	**0.015**
**Total grains**	6.75 (3.03)	6.96 (4.57)	0.20	0.578	6.69 (2.74)	6.75 (4.23)	0.07	0.902	6.82 (3.32)	7.17 (4.92)	0.35	0.501
**Whole grains**	1.16 (1.69)	1.54 (2.64)	0.38	0.069	0.96 (0.86)	1.55 (3.14)	0.58	0.099	1.36 (2.22)	1.54 (2.02)	0.18	0.430
**Plant protein**	1.16 (1.39)	1.29 (1.79)	0.13	0.431	1.12 (1.34)	1.32 (1.97)	0.20	0.400	1.20 (1.45)	1.27 (1.60)	0.05	0.811
**Plant protein** **intention**	62.06 (27.92)	64.42 (26.49)	2.36	0.222	61.34 (26.56)	62.72 (26.75)	1.38	0.623	62.77 (29.37)	66.11 (26.28)	3.34	0.214
**Animal protein intention**	75.22 (26.15)	65.17 (29.33)	**−10.05**	**---**	75.68 (26.19)	63.55 (27.84)	**−12.13**	**---**	74.76 (26.28)	66.83 (30.87)	**−7.92**	**0.013**
**Fruits, veg,** **intention**	85.30 (18.35)	87.50 (15.42)	**2.21**	**0.050**	85.51 (18.19)	88.33 (14.53)	2.81	0.098	85.08 (18.62)	86.67 (16.31)	1.60	0.283

**Table 3 foods-10-03147-t003:** Correlations between education, income, and changes in study outcomes and gender differences between changes in study outcomes (Δ indicates change in females—change in males; thus, positive numbers indicate greater changes in females). Bold values indicate *p* < 0.05.

	Education		Income		Gender	
	*r*	*p*	*r*	*p*	Δ	*p*
Values	−0.055	0.487	0.003	0.970	−0.130	0.425
Moral satisfaction	0.109	0.171	0.153	0.053	−0.266	0.199
Perceived benefits	0.045	0.576	0.098	0.221	−0.279	0.108
Perceived susceptibility	0.091	0.256	**0.157**	**0.048**	−0.152	0.567
Perceived severity	0.083	0.296	−0.094	0.239	−0.139	0.405
Subjective norms	0.145	0.068	0.040	0.620	−0.079	0.689
Self-efficacy	0.061	0.441	−0.047	0.557	−0.086	0.600
Total protein	0.183	0.021	−0.042	0.599	−0.318	0.687
Meat, poultry, and seafood	**0.202**	**0.011**	−0.006	0.942	−0.460	0.477
Meat	0.128	0.109	0.033	0.677	−0.170	0.715
Poultry	0.067	0.402	−0.087	0.276	−0.860	0.062
Seafood	−0.030	0.710	0.013	0.874	**0.831**	**0.028**
Eggs	0.151	0.058	0.020	0.807	**−0.497**	**0.047**
Soy	−0.020	0.801	−0.056	0.481	0.174	0.436
Nuts and seeds	0.011	0.893	−0.094	0.238	0.314	0.073
Legumes	−0.059	0.462	**0.224**	**0.005**	−0.455	0.083
Total dairy	−0.091	0.253	0.097	0.223	−0.124	0.643
Total fruits	0.079	0.320	−0.031	0.698	0.095	0.576
Total vegetables	0.056	0.480	−0.002	0.980	−0.090	0.662
Total grains	0.070	0.379	0.142	0.074	−1.850	0.067
Whole grains	0.009	0.914	0.043	0.587	−0.697	0.268
Plant protein	−0.044	0.584	0.061	0.446	0.025	0.954
Plant protein intention	−0.155	0.052	−0.093	0.243	−0.226	0.957
Animal food intention	−0.056	0.488	**−0.161**	**0.043**	7.563	0.124
Fruits and vegetables intention	0.008	0.918	0.021	0.795	−0.942	0.700

**Table 4 foods-10-03147-t004:** Two independent analyses were conducted to explore relationships among predictor and outcome variables. First, post-intervention variables were individually regressed on baseline predictor variables and group (health vs. environmental text messages). Standardized beta values are displayed. Second, correlations between changes in the predictor variables and changes outcome variables were explored using a Pearson correlation. Bold values indicate *p* < 0.05. Dashed lines indicate *p* < 0.001.

Predictor	Outcome	B	*p*	B (Group)	*p* (Group)	*r*	*p* (Corr.)
Values	Meat, poultry, and seafood	**0.210**	**0.009**	0.076	0.341	−0.011	0.891
	Meat	−0.018	0.831	0.046	0.590	0.119	0.134
	Fruits	0.100	0.194	0.030	0.692	0.030	0.704
	Vegetables	0.097	0.233	0.164	0.045	0.069	0.385
	Plant protein	0.071	0.409	0.006	0.940	0.090	0.258
	Plant protein intention	0.076	0.305	0.079	0.262	0.128	0.109
	Animal food intention	**−0.169**	**0.027**	−0.002	0.978	−0.072	0.372
	Fruits and vegetables intention	0.069	0.291	−0.019	0.766	0.033	0.683
	Perceived benefits	0.063	0.341	0.014	0.834	**0.176**	**0.026**
	Perceived susceptibility	**0.185**	**0.010**	−0.005	0.936	0.026	0.749
	Perceived severity	**0.255**	**---**	**−0.162**	**0.015**	0.103	0.196
	Subjective norms	**0.148**	**0.036**	−0.020	0.754	**0.166**	**0.037**
	Self-efficacy	**0.175**	**0.018**	0.022	0.740	0.139	0.081
	Moral satisfaction	**0.181**	**0.007**	0.093	0.149	0.084	0.291
Perceived benefits	Meat, poultry, and seafood	0.019	0.804	−0.006	0.941	−0.007	0.935
	Meat	−0.056	0.483	0.055	0.484	−0.053	0.509
	Fruits	0.099	0.163	−0.012	0.868	−0.090	0.257
	Vegetables	0.040	0.604	0.125	0.098	−0.026	0.744
	Plant protein	**0.155**	**0.048**	−0.027	0.728	0.015	0.856
	Plant protein intention	0.091	0.158	0.046	0.469	**0.179**	**0.024**
	Animal food intention	−0.049	0.489	0.066	0.345	−0.082	0.305
	Fruits and vegetables intention	−0.084	0.161	−0.043	0.473	0.064	0.423
Perceived susceptibility	Meat, poultry, and seafood	**0.156**	**0.037**	0.004	0.959	−0.105	0.188
	Meat	0.111	0.162	0.060	0.446	−0.118	0.138
	Fruits	−0.009	0.902	−0.009	0.902	−0.004	0.961
	Vegetables	−0.091	0.233	0.123	0.105	0.124	0.119
	Plant protein	0.022	0.779	−0.020	0.796	0.090	0.262
	Plant protein intention	**0.140**	**0.030**	0.059	0.356	0.007	0.931
	Animal food intention	−0.070	0.330	0.059	0.396	−0.107	0.183
	Fruits and vegetables intention	−0.057	0.342	−0.049	0.410	0.027	0.740
Perceived severity	Meat, poultry, seafood	0.093	0.292	0.045	0.610	−0.050	0.533
	Meat	0.029	0.759	0.068	0.464	−0.057	0.477
	Fruits	0.059	0.485	0.023	0.781	0.033	0.676
	Vegetables	−0.004	0.963	0.125	0.163	0.100	0.211
	Plant protein	0.051	0.585	0.006	0.952	0.103	0.196
	Plant protein intention	**0.185**	**0.015**	**0.149**	**0.048**	−0.069	0.390
	Animal food intention	0.034	0.687	0.083	0.319	**−0.164**	**0.040**
	Fruits and vegetables intention	0.041	0.562	−0.024	0.740	−0.089	0.263
Subjective norms	Meat, poultry, and seafood	**0.164**	**0.028**	0.003	0.967	−0.049	0.541
	Meat	0.119	0.129	0.060	0.445	−0.061	0.443
	Fruits	0.027	0.700	−0.007	0.925	−0.049	0.541
	Vegetables	0.100	0.185	0.133	0.078	0.008	0.925
	Plant protein	0.069	0.386	−0.017	0.827	0.082	0.302
	Plant protein intention	**0.189**	**0.004**	0.062	0.325	0.009	0.909
	Animal food intention	−0.150	0.038	0.054	0.438	−0.062	0.438
	Fruits and vegetables intention	−0.068	0.259	−0.050	0.405	0.001	0.989
Self-efficacy	Meat, poultry, seafood	0.103	0.170	0.007	0.928	−0.043	0.593
	Meat	−0.003	0.970	0.053	0.507	**−0.171**	**0.031**
	Fruits	0.099	0.164	0.003	0.961	−0.056	0.482
	Vegetables	0.053	0.493	0.133	0.081	0.029	0.718
	Plant protein	**0.165**	**0.036**	−0.001	0.985	<0.001	0.997
	Plant protein intention	**0.214**	**0.001**	0.077	0.220	0.118	0.139
	Animal food intention	−0.041	0.572	0.060	0.399	**−0.241**	**0.002**
	Fruits and vegetables intention	0.002	0.974	−0.046	0.450	0.092	0.250
Moral satisfaction	Meat, poultry, and seafood	−0.051	0.500	−0.002	0.981	0.001	0.993
	Meat	**−0.184**	**0.021**	0.061	0.432	−0.118	0.140
	Fruits	0.077	0.279	−0.011	0.872	−0.037	0.640
	Vegetables	0.028	0.714	0.126	0.096	−0.029	0.714
	Plant protein	0.123	0.117	−0.026	0.739	−0.069	0.388
	Plant protein intention	**0.147**	**0.023**	0.045	0.479	0.083	0.298
	Animal food intention	**−0.182**	**0.010**	0.070	0.309	0.019	0.817
	Fruits and vegetables intention	0.005	0.933	−0.046	0.443	−0.049	0.539

## Data Availability

Data can be found at the following location: doi:10.17632/fwdssw4k3p.1. Additional data are available upon reasonable request.

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
