# Peer review of "Health- or Environment-Focused Text Messages as a Potential Strategy to Increase Plant-Based Eating among Young Adults: An Exploratory Study"

_foods, 2021, doi:10.3390/foods10123147_

Round 1

Reviewer 1 Report

This is very interesting topic and paper but some parts should be improved.

  1. There are too many keywords, especially Theory of Planned Behavior and Health Belief Model that are not mentioned in Summary.
  2. Last sentence in first paragraph of Introduction section " Considering the rising interest in environmental sustainability [9,10], environ- 39
    ment-focused messaging may represent a novel strategy to improve diet quality" is unrelated with previous few sentences. Authors should explain more sustainability as factor that influence food choice.
  3. In Theoretical Framework both theories, especially HBM should be better explained (e.g. with figures and all variables in model)
  4. Why didn't you use SEM for models (TPB and HBM) testing? 
  5. Can you discuss these results " Higher education correlated with increased meat, poultry and seafood intake. Income was positively correlated with changes in legume intake and negatively correlated 
    with changes in intention to consume animal foods. Females consumed more seafood while males consumed more eggs?
  6. Table 4 is difficult to understand
  7. Conclusion should be add because discussion is long and complex

Reviewer 2 Report

The research objective is interesting. It is logically presented and supported with relevant evidences. I really appreciate for the well research design as well as the proposed implications. However, the quality of the manuscript would be improved if the authors further consider the followings:

  1. The statements in Lines 95-99 should be further refined to provide a good rationale for the sample selection used in this study. It is because several studies have already addressed the impacts of the age, marriage, etc. on the food selection.
  2. The reliability & validity of the data collection mechanism should be further clarified.
  3. The statements in Sections 4.2.2 and 4.2.3 should be supported by specific figures presented Table 4.

Author Response

Thank you for your thoughtful comments. Please see the attachment.
